# Polygenic risk score validation using Korean genomes of 265 early-onset acute myocardial infarction patients and 636 healthy controls

**Youngjune Bhak**[1,2☯], **Yeonsu Jeon**[1,2☯], **Sungwon Jeon**[1,2☯], **Changhan Yoon**[1,2], **Min Kim**[1,2], **Asta Blazyte**[1,2], **Yeonkyung Kim**[1], **Younghui Kang**[1], **Changjae Kim**[3], **Sang Yeub Lee**[4], **Jang-Whan Bae**[4], **Weon Kim**[5], **Yeo Jin Kim**[1], **Jungae Shim**[1], **Nayeong Kim**[1], **Sung Chun**[6,7], **Byoung-Chul Kim**[3], **Byung Chul Kim**[3], **Semin Lee**[1,2], **Jong Bhak**[1,2,3,8]*, **Eun-Seok Shin**[8,9]*

1 Korean Genomics Center (KOGIC), Ulsan National Institute of Science and Technology (UNIST), Ulsan, Republic of Korea, 2 Department of Biomedical Engineering, School of Life Sciences, Ulsan National Institute of Science and Technology (UNIST), Ulsan, Republic of Korea, 3 Clinomics Inc, Ulsan, Republic of Korea, 4 Division of Cardiology, Department of Internal Medicine, Chungbuk National University, College of Medicine, Cheongju, Republic of Korea, 5 Division of Cardiology, Department of Internal Medicine, Kyung Hee University Hospital, Seoul, Republic of Korea, 6 Division of Pulmonary Medicine, Boston Children's Hospital, Boston, Massachusetts, United States of America, 7 Department of Pediatrics, Harvard Medical School, Boston, Massachusetts, United States of America, 8 Personal Genomics Institute, Genome Research Foundation, Ulsan, Republic of Korea, 9 Division of Cardiology, Department of Internal Medicine, Ulsan Medical Center, Ulsan, Republic of Korea

☯ These authors contributed equally to this work.
* jongbhak@genomics.org (JB); sesim1989@gmail.com (ESS)

**Data Availability Statement:** The data presented is under legal restriction since data contain potentially

## Abstract

### Background

The polygenic risk score (PRS) developed for coronary artery disease (CAD) is known to be effective for classifying patients with CAD and predicting subsequent events. However, the PRS was developed mainly based on the analysis of Caucasian genomes and has not been validated for East Asians. We aimed to evaluate the PRS in the genomes of Korean early-onset AMI patients (n = 265, age ≤50 years) following PCI and controls (n = 636) to examine whether the PRS improves risk prediction beyond conventional risk factors.

### Results

The odds ratio of the PRS was 1.83 (95% confidence interval [CI]: 1.69–1.99) for early-onset AMI patients compared with the controls. For the classification of patients, the area under the curve (AUC) for the combined model with the six conventional risk factors (diabetes mellitus, family history of CAD, hypertension, body mass index, hypercholesterolemia, and current smoking) and PRS was 0.92 (95% CI: 0.90–0.94) while that for the six conventional risk factors was 0.91 (95% CI: 0.85–0.93). Although the AUC for PRS alone was 0.65 (95% CI: 0.61–0.69), adding the PRS to the six conventional risk factors significantly improved the accuracy of the prediction model ($P = 0.015$). Patients with the upper 50% of PRS showed a higher frequency of repeat revascularization (hazard ratio = 2.19, 95% CI: 1.47–3.26) than the others.

identifying or sensitive patient information. Therefore, raw sequencing data, individual genotype information, and clinical trait data will be available upon request and after an approval from the Korean Genomics Center's review board in UNIST. Information about the KGP, present dataset (Cardiomics) and other related data sharing can be found at http://koreangenome.org/Cardiomics.

**Funding:** This work was supported by the U-K BRAND Research Fund (1.190007.01) of UNIST; Research Project Funded by the U-K BRAND Research Fund (1.200108.01) of UNIST; Research Project Funded by Ulsan City Research Fund (1.190033.01) of UNIST; Research Project Funded by Ulsan City Research Fund (1.200047.01) of UNIST; Research Project Funded by Ulsan City Research Fund (2.180016.01) of UNIST. This work was also supported by the Technology Innovation Program (20003641, Development and Dissemination on National Standard Reference Data) funded by the Ministry of Trade, Industry & Energy (MOTIE, Korea). This work was also supported by internal funding of Clinomics Inc. The funder provided support in the form of salaries for authors C.K., B.K., B.C.K., and J.B., but did not have any additional role in the study design, data collection and analysis, decision to publish, or preparation of the manuscript. The specific roles of these authors are articulated in the 'author contributions' section.

**Competing interests:** C.K. and B.K. are employees, and B.C.K. and J.B. are the co-CEOs of Clinomics Inc. B.C.K. and J.B. have an equity interest in the company. This does not alter our adherence to PLOS ONE policies on sharing data and materials. All other authors have no conflict of interest to declare.

## Conclusions

The PRS using 265 early-onset AMI genomes showed improvement in the identification of patients in the Korean population and showed potential for genomic screening in early life to complement conventional risk prediction.

## Introduction

The polygenic risk score (PRS) is a quantitative genetic risk score, determined by the cumulative impact of genome-wide variants, used to improve risk prediction for common chronic diseases [1]. A study of the PRS for coronary artery disease (CAD) reported a significant improvement in classification when the PRS was combined with conventional risk factors [2]. The study also reported the more efficient classification of patients using the PRS in a younger age group (age <55 years) than in an older age group (age ≥55 years). The PRS also showed predictive power for all-cause mortality after cardiac catheterization [3].

In 2019, a study on early-onset myocardial infarction (mean age of the patients = 48 years) revealed 10-fold higher classification capacity of the PRS compared to a classification based on monogenic mutations [4]. However, the study did not fully evaluate the contribution of the PRS when it was combined with conventional risk factors, such as smoking, for the classification of patients. Additionally, the proportion of high PRS carriers was insignificant in the Asian patient group. This is probably because of the small number of Asian patients (n = 40, 1.9% among patients) or the use of the PRS derived from studies performed mainly on Caucasian individuals [5].

The incidence of acute myocardial infarction (AMI) varies by ethnic group, with particularly lower values in East Asian populations than Western populations [6,7]. This variation among ethnic groups may be caused by differences in genetic factors since East Asian and Caucasian populations are genetically distinct [8]. Therefore, validating the applicability of the PRS in a different ethnic group is critical, particularly for East Asian patients.

Herein, we applied the whole-genome sequencing-based PRS in 265 Korean early-onset AMI patients following percutaneous coronary intervention (PCI). We evaluated the validity of the PRS in Korean patients with early-onset AMI in terms of the classification of patients and the prediction of cardiovascular events after PCI.

## Materials and methods

### Study population

We obtained the Korean variome and clinical information data from KGP. The KGP is a joint project facilitated by the Personal Genome Project (PGP) at Harvard Medical School, National Center for Standard Reference Data of Korea, Clinomics, Inc., and KOGIC (Korean Genomics Center) of UNIST. It aims to generate a combination of whole-genome sequencing data, questionnaires, and clinical measurements of participants in Korea. The Korean patients were hospitalized with a diagnosis of and treatment for an ST-segment elevation myocardial infarction or non-ST-segment elevation myocardial infarction; they were ≤50 years old and had undergone PCI at three hospitals. The Korean control subjects were selected from among the KGP individuals without a history of AMI, angina, or heart attack. Subjects who were taking drugs for CAD were excluded. Written informed consent was obtained from all study participants by the clinicians in the participating hospitals. The present study was approved by the UNIST

Institutional Review Board (UNISTIRB-15-19-A) and was performed in accordance with the Declaration of Helsinki.

## Genomic variant identification

The Korean variome was derived from KGP. A detailed description of the sequencing and genotyping is described in the previous KGP initiation paper [8]. Briefly, the adapters were trimmed using Cutadapt (ver 1.9.1) [9]. The mapped BAM files were sorted using genomic coordination in Picard (ver. 2.14.0) with the SortSam module. Duplicated reads were marked using Picard (ver. 2.14.0) with the MarkDuplicates module. Mapping quality was calibrated using the BaseRecalibrator module in the Genome Analysis Tool Kit (ver. 3.7) [10]. Joint variant genotyping was performed using HaplotypeCaller in the Genome Analysis Tool Kit with the '-stand_call_conf 30' option. To extract variants in the callable genomic region, variants were filtered based on strict accessible regions as defined by the 1,000 Genome Project [11].

## PRS calculation

We calculated the PRSs of the patients and controls based on the reported list of allele variants and their risk weights for CAD [1]. Briefly, this risk prediction model was originally derived by running the LDpred algorithm on the estimated genetic effects from a meta-analysis of CAD [5,12]. The acquired variome from KGP was lifted-over to hg19 by CrossMap (ver 0.2.7), and the PRS was calculated using PLINK (ver 1.90) with the "—score" option [13,14]. Downstream analyses were performed using R version 3.6.3 software [15]. The calculated PRS was normalized by inverse normal transformation.

## Patient follow-up and outcome measurements

We conducted the follow-up of patients at outpatient clinic visits and through telephonic contact. An independent clinical event committee whose members were blinded to the clinical, angiographic, and genetic data adjudicated all events. The vital status of all patients was cross-checked using Korean Health System's unique identification numbers. In this way, mortality was confirmed, even in patients who were lost during follow-up. The adverse events included all causes of death, MI, and repeat revascularization. All clinical outcomes were defined according to the Academic Research Consortium [16].

## Statistical analysis

Downstream analyses were performed using R version 3.6.3 software [15]. The calculated PRSs were standardized to zero-mean and one-standard deviation by inverse normal transformation. The distribution of the PRSs was compared between patients and controls using the Wilcoxon ranksum test. The correlation between the PRS and age was assessed using Spearman's rank correlation. A high PRS in the distribution comparison analysis was defined as a PRS higher than the top 5% of the control distribution [1,4]. Receiver operating characteristic curve analysis was conducted using the R package pROC (ver 1.16.1) [17]. The paired test was conducted for the comparison of the areas under the curve (AUCs) among predictors. The PRS was standardized among patients, and the patients were divided into an upper-50% PRS group and a lower-50% PRS group for survival analysis. Survival analysis was conducted using R package survival (ver 3.1–11) [18].

**Table 1. Baseline characteristics of patients with early-onset AMI and controls.**

| Variables | Early-onset AMI (n = 265) | Control (n = 636) |
|---|---|---|
| Male | 252 (95.1) | 323 (50.8) |
| Age, years | 46 (42–46) | 43 (29–57) |
| Body mass index | 25.5 ± 3.8 | 24.0 ± 3.5 |
| Hypertension | 78 (30.8) | 93 (14.6) |
| Diabetes mellitus | 38 (15.0) | 35 (5.5) |
| Current smoking | 178 (72.7) | 82 (12.9) |
| Hypercholesterolemia | 230 (86.8) | 294 (46.2) |
| Family history of CAD | 41 (16.4) | 21 (3.3) |
| Lipid levels, mg/dL | | |
| Total cholesterol, mg/dl | 205.8 ± 47.6 | 179.9 ± 34.2 |
| LDL cholesterol, mg/dl | 127.5 ± 43.1 | 116.0 ± 33.0 |
| HDL cholesterol, mg/dl | 42.7 ± 12.9 | 57.4 ± 13.9 |
| Triglycerides, mg/dl | 199.5 ± 147.2 | 116.0 ± 77.1 |

AMI, acute myocardial infarction; family history of CAD, 1st degree family history of coronary artery disease; LDL, low-density lipoprotein; HDL, high-density lipoprotein. Values are mean ± SD, median (interquartile range, 25th—75th), or n (%).

## Results

### Sample characteristics

A total of 901 whole-genomes from 265 patients with early-onset AMI ($\leq$50 years old, number of male patients: 252, number of female patients: 13), and 636 controls (see Methods; Table 1) were sequenced and analysed. The mean age of the patients and controls was 44.6 years (median = 46, interquartile range [IQR]: 42 to 46) and 43.8 years (median = 43, IQR: 29 to 57), respectively. The median follow-up period in the patient group was 43 months (IQR: 16 days to 14.8 years). The proportions of current smokers in the patient and control groups were 72.7% and 12.9%, respectively. A total of 75.1% of the male patients and 22.6% of the male controls were current smokers.

### Differences in the PRS between the patients and controls

The distribution of PRSs was significantly higher in patients than in controls (average PRSs were 0.40 and -0.17 for patients and controls, respectively; $P$ <0.001). The odds ratio of the PRS for early-onset AMI patients compared with the controls was 1.83 (95% confidence interval [CI]: 1.69–1.99, $P$ <0.001). The proportion of individuals who show a high PRS which was defined as a PRS higher than the top 5% in the control distribution was significantly larger among the patients (58 of 265, 21.9%) than among the controls (32 of 636, 5.0%, $P$<0.001). In the patients group, the PRS and age (age-at-event) showed a significantly negative correlation (Spearman's rho = -0.14, $P$ = 0.025), while not significant correlation in the control group (Spearman's rho = 0.03, $P$ = 0.463).

### Classification power of PRS for patient classification

The AUC for the PRS was 0.65 (95% CI: 0.61–0.69). The AUC for the classification model including all six conventional risk factors was 0.91 (95% CI: 0.89–0.93) and that of the classification model including the six conventional risk factors and the PRS was 0.92 (95% CI: 0.90–0.94). The contribution of the PRS to the six conventional risk factors was significant ($P$ = 0.015) (Fig 1). Among conventional risk factors, current smoking showed the highest

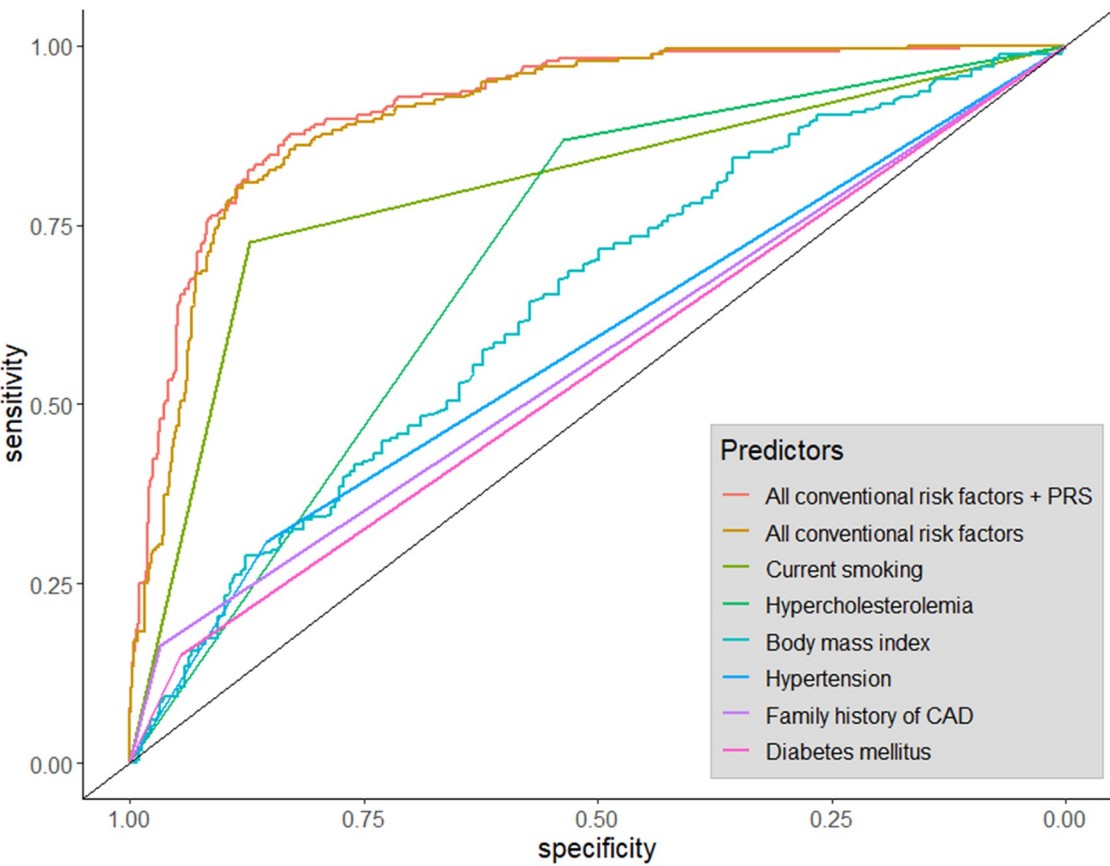

| Predictors | AUC | 95% CI | *P* for AUC increment |
|---|---|---|---|
| All conventional risk factors + PRS | 0.92 | 0.90 – 0.94 | 0.015 (vs All conventional risk factors) |
| All conventional risk factors | 0.91 | 0.89 – 0.93 | <0.001 (vs Current smoking) |
| Current smoking | 0.80 | 0.77 – 0.83 | <0.001 (vs Hypercholesterolemia) |
| Hypercholesterolemia | 0.70 | 0.68 – 0.73 | <0.001 (vs Body mass index) |
| Body mass index | 0.64 | 0.60 – 0.68 | 0.043 (vs Hypertension) |
| Hypertension | 0.58 | 0.55 – 0.61 | 0.396 (vs Family history of CAD) |
| Family history of CAD | 0.57 | 0.54 – 0.59 | 0.334 (vs Diabetes mellitus) |
| Diabetes mellitus | 0.55 | 0.52 – 0.57 | <0.001 (vs baseline) |

**Fig 1. Receiver operator characteristic curve and AUC for conventional risk factors and combined models.** AUC, area under the curve; CAD: coronary artery disease; CI: confidence interval; PRS, polygenic risk score.

AUC of 0.80 (95% CI: 0.77–0.83) compared to other factors such as hypercholesterolemia (AUC = 0.70, 95% CI: 0.68–0.73), body mass index (0.64, 95% CI: 0.60–0.68), hypertension (0.58, 95% CI: 0.55–0.61), family history of CAD (0.57, 95% CI: 0.54–0.59), and diabetes mellitus (0.55, 95% CI: 0.52–0.57).

The AUC for the PRS was significantly higher in the younger age group (AUC = 0.69, 95% CI: 0.63–0.75) than in the older age group (AUC = 0.58, 95% CI: 0.50–0.66) (*P* = 0.029) when we compared the classification accuracy of the PRS between younger subjects (25 < age ≤ 45 years, 130 patients and 248 controls) and older subjects (45 < age ≤ 50 years, 134 patients and 72 controls) (Fig 2). Combining the PRS with the conventional risk factors increased the classification accuracy in both the younger (AUC of conventional factors = 0.92, 95% CI: 0.90–0.95;

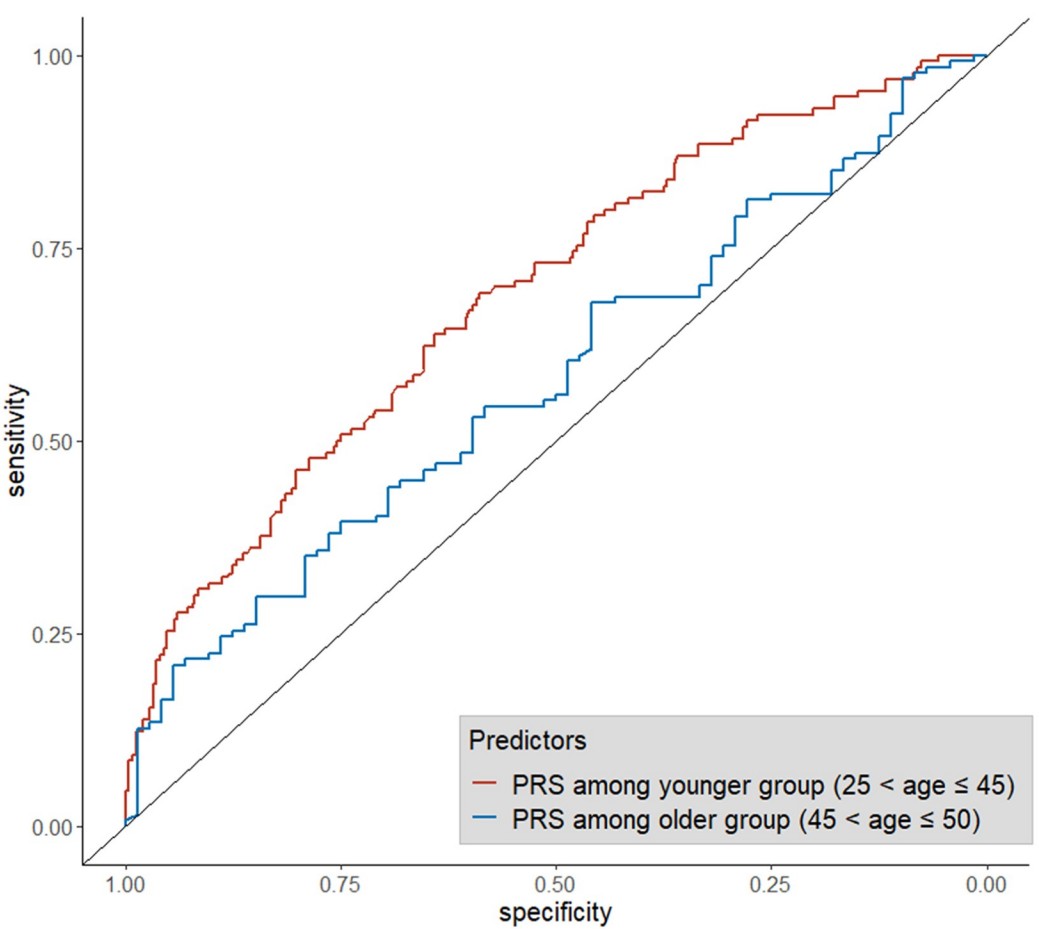

| Predictors | AUC | 95% CI | *P* for AUC increment |
|---|---|---|---|
| PRS among younger group | 0.69 | 0.63 – 0.75 | 0.015 (vs PRS among older group) |
| PRS among older group | 0.58 | 0.50 – 0.66 | <0.001 (vs baseline) |

**Fig 2. Receiver operating characteristic curve and AUC for PRS stratified by age.** AUC, area under the curve; CI: confidence interval; PRS, polygenic risk score.

AUC of conventional factors and PRS = 0.94, 95% CI: 0.91–0.96; *P* = 0.038) and the older groups (AUC of the conventional factors: 0.90, 95% CI: 0.86–0.95; AUC of the conventional factors and PRS: 0.91, 95% CI: 0.86–0.95; *P* = 0.423). However, the additional improvment in discrimination accuracy was significant only in the younger group.

## Prediction power of the PRS for subsequent cardiovascular events

A significant cumulative event was only identified for repeat revascularization when we assessed the classification power of the PRS for predicting a subsequent cardiovascular event after PCI. The cumulative event of all causes of death or AMI was not significant (all causes of death *P* = 0.944, AMI *P* = 0.957), possibly because of the small sample size and the small number of events (all causes of death: n = 5, AMI: n = 4). Patients with upper 50% PRS among patients showed a significantly higher frequency of repeat revascularization (hazard ratio = 2.19, 95% CI: 1.47–3.26, *P* = 0.049; Fig 3). PRS was the only variable that was

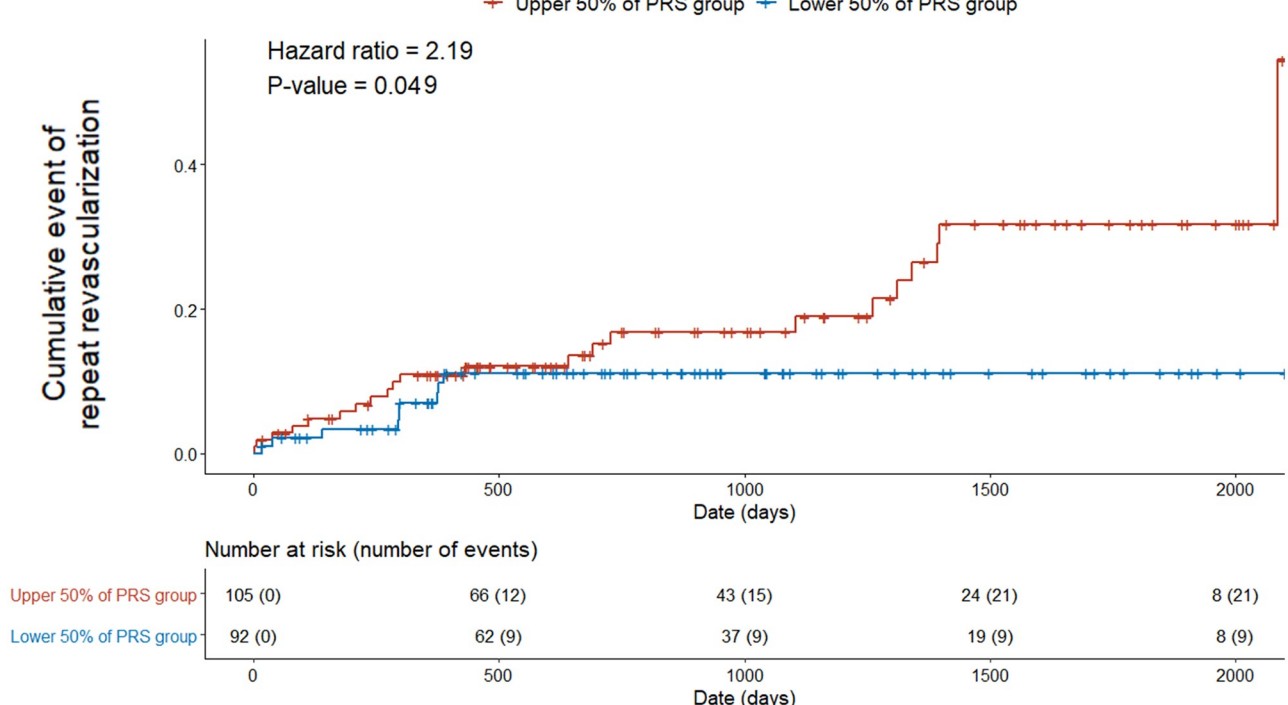

**Fig 3. Comparison of the cumulative incidence of repeat revascularization events between the upper-50% and lower-50% PRS groups.** Number at risk, the number of followed individuals; number of events, the number of individuals with repeat revascularization; PRS, polygenic risk score.

significantly and independently associated with the cumulative event of repeat revascularization after PCI in both univariable and multivariable analyses conducted with the inclusion of all the conventional risk factors (Table 2).

## Discussion and conclusions

The PRS distribution was significantly higher in patients than in the controls when we evaluated the discrimination and prediction power of the PRS for CAD in Korean early-onset AMI patients following PCI. The odds ratio of the PRS was significantly higher (1.83) in the patient group. The proportion of high PRS carriers was also significantly higher in patients than in

**Table 2. Predictive power of conventional risk factors and PRS for repeat revascularization after PCI.**

| Predictors | Univariable analysis | | | Multivariable analysis | | |
|---|---|---|---|---|---|---|
| | Hazard ratio | 95% CI | *P*-value | Hazard ratio | 95% CI | *P*-value |
| **Body mass index** | 0.96 | 0.88–1.04 | 0.314 | 0.92 | 0.83–1.03 | 0.135 |
| **Hypertension** | 0.74 | 0.32–1.75 | 0.495 | 0.88 | 0.36–2.18 | 0.787 |
| **Current smoking** | 0.67 | 0.31–1.45 | 0.313 | 0.58 | 0.26–1.31 | 0.191 |
| **Diabetes mellitus** | 1.44 | 0.59–3.55 | 0.424 | 1.41 | 0.54–3.69 | 0.478 |
| **Hypercholesterolemia** | 3.33 | 0.45–24.50 | 0.238 | 3.56 | 0.43–29.40 | 0.238 |
| **Family history of CAD** | 0.77 | 0.27–2.20 | 0.621 | 0.63 | 0.21–1.91 | 0.411 |
| **Polygenic risk score** | 1.64 | 1.12–2.38 | **0.010** | 1.65 | 1.11–2.46 | **0.014** |

CI, confidence interval; Family history of CAD, 1st degree family history of coronary artery disease; Predictors in the multivariable analysis included all conventional risk factors in the univariable analysis.

controls. This seemed to indicate a significant increase in discrimination accuracy when it was combined with conventional risk factors. We also observed a significantly higher frequency of repeat revascularization events in the patient group with PRSs that fell within the upper 50% than in the patient group with those that fell within the lower 50%. This suggests that the PRS can be a useful indicator as genetic screening which make AMI patients notice the possiblity of repeat occurrences of revascularization after PCI. Our investigation indicates that the PRS is also applicable to Korean early-onset AMI patients.

The AUC for current smoking status was the highest predictor in the conventional predictors. One possible issue is that our patient cohort contained a higher proportion of current smokers (patients: 72.7%, controls: 12.9%) than a previous study on early-onset AMI (patients: 51%, controls: 12%) [4]. This difference in the proportions of smokers between the previous study and the present study could be due to the difference in the proportion of males among patients between the previous (34%) and present studies (95.1%). Furthermore, the proportion of current smokers among both male patients (75.1%) and male controls (22.6%) in the present study was not similar to the reported proportion of smokers among Korean males (40 to 50%) [19]. The low proportion of current smokers among the controls could have introduced a strong bias in our analyses. Another sampling bias is the way in which the control individuals were recruited. The controls were healthy volunteers from the Korean Genome Project (KGP) who are probably interested in maintaining good healthOverall, the high AUC for current smoking status in the present study could have been biased. The classification accuracy based on the current smoking status may be lower in practice. On the other hand, the sample recruitment bias could have caused the underestimation of PRS accuracy.

Nevertheless, PRS measurement, if cost and time for sequencing are considered, can be performed when conventional risk factors cannot be determined. In this context, identifying genetic risk at birth or in early life would be the earliest and the most cost-effective option. The negative correlation between age-at-event and the PRS of the patient group and the higher discrimination accuracy of the PRS in the younger group than in the older group indicate an age-dependent difference in the weight of genetic effects, at least for early-onset AMI among Koreans. This suggests that adjusting its effect weight depending on age can improve the contribution of the PRS to the conventional model, and thus, as one becomes older, early-assessement of PRSs for AMI can be combined with periodically measured conventional risk factors to stratify individuals who have different trajectories of AMI risk and predict the early events associated with AMI. Hence, the prediction of the risk trajectory with the age-adjusted PRS weighing model can be beneficial, especially in young individuals, because the geneticrisk could be attenuated by adhering to a healthy lifestyle as early as possible [20,21].

For example, it has been reported that significant coronary atherosclerosis already exists in young and asymptomatic people [22]. As the number of cardiovascular risk factors increases, so does the severity of asymptomatic coronary atherosclerosis in young people [23]. If the risk of CAD in a younger person can be determined before noticing any cardiovascular risk factors, early risk modification and prevention of asymptomatic coronary atherosclerosis can be achieved. By doing so, it would be possible to reduce the prevalence of CAD and achieve primary prevention. Thus, the PRS can serve as a guide to achieving primary prevention.

We found that the association of the PRS with repeat revascularization events after PCI was significant, but the association with repeat revascularization was insignificant for conventional risk factors. This may suggest that the PRS can better explain the possibility of repeat revascularization and be a practical measure for guiding secondary prevention strategies. For example, after PCI, a clinician may recommend patients with a high PRS to visit a hospital more frequently than those with a low PRS. Therefore, closer follow-ups with optimal medical treatments in high PRS groups would be recommended. And such a PRS application for the

follow-up and treatment will possibly become more effective and precise if the confounding effect of clinical factors such as detailed information of diseases, subgroups of disease, drugs taking, method of treatment, kind of outcomes, durations through onset, treatment, discharge, and follow-up are considered together.

The classification accuracy of the PRS alone and the additive contribution of the PRS to the six conventional risk factors were modest; this was investigated on a previous PRS study [2]. The performance of the PRS could be affected by various factors such as the population ethnicities, disease types, and biological pathways for the construction and application of the PRS [24–28]. This indicates that improved applicability may be expected if the PRS is fine-tuned to such factors.

In conclusion, we found that the Caucasian population-based PRS is applicable to Korean patients with early-onset AMI. The PRS improved the classification accuracy of the conventianl factors for early-onset AMI with a statistical significance, although the amount of improvement was modest. The PRS was a independent factor to predict future repeat revascularization events after PCI.

## Acknowledgments

We appreciate all participants and Ulsan citizens to support Genome Korea in Ulsan project which provided Korea10K genome information. The biospecimens for this study were provided by Ulsan Medical Center and the Biobanks of Chungbuk National University Hospital (18–27, 20–04), Kyung Hee University Hospital (2018–4, 2019–4, 2019–6), and Ulsan University Hospital (60SA2017002-005) the members of the National Biobank of Korea, which is supported by the Ministry of Health, Welfare and Family Affairs. We thank the Korea Institute of Science and Technology Information (KISTI) provided us the Korea Research Environment Open NETwork (KREONET).

## Author Contributions

**Conceptualization:** Youngjune Bhak, Byoung-Chul Kim, Byung Chul Kim, Jong Bhak, Eun-Seok Shin.

**Data curation:** Yeo Jin Kim, Jungae Shim, Nayeong Kim, Eun-Seok Shin.

**Formal analysis:** Youngjune Bhak, Yeonsu Jeon, Sungwon Jeon.

**Funding acquisition:** Semin Lee.

**Investigation:** Youngjune Bhak, Yeonsu Jeon, Sungwon Jeon.

**Methodology:** Youngjune Bhak, Eun-Seok Shin.

**Project administration:** Eun-Seok Shin.

**Resources:** Yeonkyung Kim, Younghui Kang, Changjae Kim, Sang Yeub Lee, Jang-Whan Bae, Weon Kim, Jungae Shim, Nayeong Kim, Eun-Seok Shin.

**Software:** Youngjune Bhak.

**Supervision:** Eun-Seok Shin.

**Visualization:** Youngjune Bhak.

**Writing – original draft:** Youngjune Bhak.

**Writing – review & editing:** Youngjune Bhak, Yeonsu Jeon, Sungwon Jeon, Changhan Yoon, Min Kim, Asta Blazyte, Sung Chun, Semin Lee, Jong Bhak, Eun-Seok Shin.

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
