## [Decision Letter · Decision Letter 0]

26 Nov 2020

PONE-D-20-27402

Polygenic risk score validation using Korean genomes of 265 early-onset acute myocardial infarction patients and 636 healthy controls

PLOS ONE

Dear Dr. Bhak,

Thank you for submitting your manuscript to PLOS ONE. After careful consideration, we feel that it has merit but does not fully meet PLOS ONE’s publication criteria as it currently stands. Therefore, we invite you to submit a revised version of the manuscript that addresses the points raised during the review process.

We look forward to receiving your revised manuscript.

Kind regards,

Yiqiang Zhan

Academic Editor

PLOS ONE

Journal Requirements:

2.We note that you have indicated that data from this study are available upon request. PLOS only allows data to be available upon request if there are legal or ethical restrictions on sharing data publicly. For information on unacceptable data access restrictions, please see http://journals.plos.org/plosone/s/data-availability#loc-unacceptable-data-access-restrictions.

4.Thank you for stating the following in the Competing Interests:

[I have read the journal's policy and the authors of this manuscript have the following competing interests: C.K. and B.K. is an employee, and B.C.K. and J.B. are the CEOs of Clinomics Inc. B.C.K. and J.B. have an equity interest in the company. All other authors have no conflicts of interest to declare.]. 

We note that one or more of the authors have an affiliation to the commercial funders of this research study : Clinomics Inc

Reviewers' comments:

Reviewer's Responses to Questions

**Comments to the Author**

1. Is the manuscript technically sound, and do the data support the conclusions?

Reviewer #1: Yes

Reviewer #2: Partly

2. Has the statistical analysis been performed appropriately and rigorously? 

Reviewer #1: Yes

Reviewer #2: No

3. Have the authors made all data underlying the findings in their manuscript fully available?

Reviewer #1: Yes

Reviewer #2: No

4. Is the manuscript presented in an intelligible fashion and written in standard English?

Reviewer #1: Yes

Reviewer #2: Yes

5. Review Comments to the Author

Reviewer #1: The authors of this study aimed to determine whether the polygenic risk score (PRS) based on the whole genomic sequencing of 265 early AMI patients would be helpful in classifying patients. They also tried to confirm whether PRS would help predict cardiovascular events after PCI procedure. In the previously published CAD-related PRS studies, PRS is known to be a significant factor in patient classification. However, as this study includes a small number of Asian patients, it is not yet known whether PRS will help classify Asian patients with AMI. This study seems to be of great value in that it tried to find out whether PRS helps classify AMI patients in Asians. However, there are some questions about the analysis, so they are pointed out below.

First, it is difficult to understand what odds ratio 1.83 is for (page 7, line 17-18). Describe more clearly what odds ratio is for the outcome.

Second, why didn't the authors put the AUC for the PRS itself in Fig 1?

Third, in the analysis of Fig. 2, is there a reason that the age classification was 45? In the case of the control group, the age distribution is from 29 to 57 years old. Is there a reason to include only 45 to 50 years old in the old age group? If the age group is set as above, only 72 control groups are included in the old age group. But don’t these settings make it seem like there are differences in statistics?

Reviewer #2: Manuscript PONE-D-20-27402 Thank you for giving us the opportunity to review the manuscript Title: “Polygenic risk score validation using Korean genomes of 265 early-onset acute myocardial infarction patients and 636 healthy controls” A manuscript in which the author described The benefit of polygenic risk score (PRS) for classifying patients with CAD and predicting further events, the PRS in early-onset AMI genomes showed improvement in the identification and genomic screening of Korean patients in early life for health risk prediction.. We have some points we would like to refer:

Comments:

-In diabetic patient the author did not identify type (I or II) and therapy (oral or insulin) as it strongly affects MACE in follow up.

-Repeat revascularization related to many technical factors either stent (DES drug type) stent size (diameter and length) etc..

-Repeat revascularization should be identified either ( target lesion , target vessel or non-target vessel ) revascularization.

-It is not clear the time of revascularization after the onset of symptoms .

-AMI should be subgroup analysis between STEMI and non- STEMI group).

-Death should be identified either cardiac death or all causes of death

-Other MACE components are beneficial in identifying the atherosclerosis progression specialy in young age group as stroke reinfarction or stent thrombosis

6. PLOS authors have the option to publish the peer review history of their article (what does this mean?). If published, this will include your full peer review and any attached files.

Reviewer #1: No

Reviewer #2: No

---

## [Author Response · Author response to Decision Letter 0]

4 Dec 2020

Journal Requirements:

We have re-checked the journal’s style requirements and updated the manuscript as follows: 

*We changed the file names of our manuscript and cover letter.

*In the author list part, we changed affiliation indication to number.

*In the author list part, we changed the symbol for equal contribution to the provided one (¶).

*In the affiliation part, we removed ZIP or postal codes.

*In the corresponding authorship part, we excluded physical addresses and left only email addresses.

*In the corresponding authorship part, we added the corresponding authors’ initials after the authors’ email addresses.

*In the manuscript, we removed the Keywords part from the title page.

*In the manuscript, we have changed the font size and headings to the PLOS ONE template.

*In the manuscript, we have changed the name of the Introduction section from Background to Introduction.

*In the manuscript, we have changed the name of the Material and methods section from Methods to Materials and methods.

*In the manuscript, we moved the Materials and methods section following after the Introduction section.

*In the manuscript, we have changed the citation of figures as Fig x.

*In the manuscript, we moved figure captions and table directly after the paragraph in which they are first cited.

*We removed the figure images from the manuscript and submitted as separate files.

*We removed the Declarations sections containing “Ethics approval and consent to participate”, “Availability of data and materials”, “Competing Interests”, “Funding”, and “Authors’ Contributions” from the manuscript.

*We converted the format of reference into Vancouver style. 

2.We note that you have indicated that data from this study are available upon request. PLOS only allows data to be available upon request if there are legal or ethical restrictions on sharing data publicly. For information on unacceptable data access restrictions, please see http://journals.plos.org/plosone/s/data-availability#loc-unacceptable-data-access-restrictions.

 � The data presented here are available upon request since there is a legal restriction on sharing data publicly in Korea. We added detailed information for this issue in the revised cover letter as below:

“The data presented is under legal restriction since data contain potentially identifying or sensitive patient information. Therefore, raw sequencing data, individual genotype information, and clinical trait data will be available upon request and after approval from the Korean Genomics Center’s review board in UNIST. Information about the KGP and other data sharing can be found at http://koreangenome.org/Cardiomics.”

We have deleted the Ethics Statement section in the Declaration section.

4.Thank you for stating the following in the Competing Interests:

[I have read the journal's policy and the authors of this manuscript have the following competing interests: C.K. and B.K. is an employee, and B.C.K. and J.B. are the CEOs of Clinomics Inc. B.C.K. and J.B. have an equity interest in the company. All other authors have no conflicts of interest to declare.]. 

We note that one or more of the authors have an affiliation to the commercial funders of this research study : Clinomics Inc

 � We have re-checked Authors’ Contribution and added the following sentences “The funder provided support in the form of salaries for authors C.K., B.K., B.C.K., and J.B., but did not have any additional role in the study design, data collection and analysis, decision to publish, or preparation of the manuscript. The specific roles of these authors are articulated in the ‘author contributions’ section.” at the end of the Funding Statement as below:

“This work was supported by the U-K BRAND Research Fund (1.190007.01) of UNIST; Research Project Funded by Ulsan City Research Fund (1.190033.01) of UNIST; Research Project Funded by Ulsan City Research Fund (1.200047.01) of UNIST; Research Project Funded by Ulsan City Research Fund (2.180016.01) of UNIST. This work was also supported by the Technology Innovation Program (20003641, Development and Dissemination on National Standard Reference Data) funded by the Ministry of Trade, Industry & Energy (MOTIE, Korea). This work was also supported by internal funding of Clinomics Inc. The funder provided support in the form of salaries for authors C.K., B.K., B.C.K., and J.B., but did not have any additional role in the study design, data collection and analysis, decision to publish, or preparation of the manuscript. The specific roles of these authors are articulated in the ‘author contributions’ section.”

 � We have re-checked the Competing Interest Statement and added the following sentences “This does not alter our adherence to PLOS ONE policies on sharing data and materials.” in the middle of the Competing Interest Statement as below:

“C.K. and B.K. are employees, and B.C.K. and J.B. are the co-CEOs of Clinomics Inc. B.C.K. and J.B. have an equity interest in the company. This does not alter our adherence to PLOS ONE policies on sharing data and materials. All other authors have no conflict of interest to declare.”

We have attached both an updated Funding Statement and Competing Interest Statement in the revised cover letter.

Reviewers' comments:

Reviewer's Responses to Questions

Comments to the Author

1. Is the manuscript technically sound, and do the data support the conclusions?

Reviewer #1: Yes

Reviewer #2: Partly

We have modified manuscript to accommodate reviewers’ suggestions. (see below point-by-point responses)

2. Has the statistical analysis been performed appropriately and rigorously?

Reviewer #1: Yes

Reviewer #2: No

We have presented responses to accommodate reviewers’ comments. (see below point-by-point responses)

 

3. Have the authors made all data underlying the findings in their manuscript fully available?

Reviewer #1: Yes

Reviewer #2: No

For the data availability issue, we have added the detailed reason for the restricted data availability with the contact point for the data request in the revised cover letter below.

“The data presented is under legal restriction since data contain potentially identifying or sensitive patient information. Therefore, raw sequencing data, individual genotype information, and clinical trait data will be available upon request and after an approval from the Korean Genomics Center’s review board in UNIST. Information about the KGP, present dataset (Cardiomics), and other related data sharing can be found at http://koreangenome.org/Cardiomics.”

4. Is the manuscript presented in an intelligible fashion and written in standard English?

Reviewer #1: Yes

Reviewer #2: Yes

5. Review Comments to the Author

Reviewer #1: The authors of this study aimed to determine whether the polygenic risk score (PRS) based on the whole genomic sequencing of 265 early AMI patients would be helpful in classifying patients. They also tried to confirm whether PRS would help predict cardiovascular events after PCI procedure. In the previously published CAD-related PRS studies, PRS is known to be a significant factor in patient classification. However, as this study includes a small number of Asian patients, it is not yet known whether PRS will help classify Asian patients with AMI. This study seems to be of great value in that it tried to find out whether PRS helps classify AMI patients in Asians. However, there are some questions about the analysis, so they are pointed out below.

First, it is difficult to understand what odds ratio 1.83 is for (page 7, line 17-18). Describe more clearly what odds ratio is for the outcome.

Thank you for pointing out the vague description on odd ratio in the manuscript. We have described the patient as “early-onset AMI (acute myocardial infarction)” to make the endpoint of comparison clear for the readers (line 6, on page 9).

Second, why didn't the authors put the AUC for the PRS itself in Fig 1?

The reason we did not put AUC for our PRS itself in Fig 1 is because we wanted to present and emphasize the contribution of the PRS to the conventional risk factor rather than showing the performance of PRS itself as a single score parameter. Therefore, we chose the current Fig 1 and placed the description for the performance of PRS itself at the start of the section rather than adding the performance to the figure.

Third, in the analysis of Fig. 2, is there a reason that the age classification was 45? In the case of the control group, the age distribution is from 29 to 57 years old. Is there a reason to include only 45 to 50 years old in the old age group? If the age group is set as above, only 72 control groups are included in the old age group. But don’t these settings make it seem like there are differences in statistics?

We used age 45 to classify/divide the samples for abtraining a most balanced number of patients between two age groups (For 25 < age ≤ 45 years, 130 patients. For 45 < age ≤ 50 years, 134 patients). Apart from that, if we consider the significant negative correlation between age and PRS in the patient group (Spearman’s rho = -0.14, P = 0.025), and the lack of significant correlation between age and PRS in the control group (Spearman’s rho = 0.03, P = 0.463), the age-dependent accuracy of PRS was expected to show the comparable tendency even in different settings as also reported from the previous study (PMID: 32068818).

 

Reviewer #2: Manuscript PONE-D-20-27402 Thank you for giving us the opportunity to review the manuscript Title: “Polygenic risk score validation using Korean genomes of 265 early-onset acute myocardial infarction patients and 636 healthy controls” A manuscript in which the author described The benefit of polygenic risk score (PRS) for classifying patients with CAD and predicting further events, the PRS in early-onset AMI genomes showed improvement in the identification and genomic screening of Korean patients in early life for health risk prediction.. We have some points we would like to refer:

Comments:

-In diabetic patient the author did not identify type (I or II) and therapy (oral or insulin) as it strongly affects MACE in follow up.

-Repeat revascularization related to many technical factors either stent (DES drug type) stent size (diameter and length) etc..

-Repeat revascularization should be identified either ( target lesion , target vessel or non-target vessel ) revascularization.

-It is not clear the time of revascularization after the onset of symptoms .

-AMI should be subgroup analysis between STEMI and non- STEMI group).

-Death should be identified either cardiac death or all causes of death

-Other MACE components are beneficial in identifying the atherosclerosis progression specialy in young age group as stroke reinfarction or stent thrombosis

Thank you for the comments. The factors you listed are known to be associated with follow-up events and insufficient information for retrospectively collected patients limited the range of present analysis. Although the previous PRS based follow-up event prediction study reported a high risk for all-cause mortality from the high PRS group with similar consideration to the present manuscript [PMID: 30571185], extended studies accompanied with detailed clinical factors that you have listed may be required through large-scale randomized control trials for the application of PRS in the real field. Therefore, we have added the sentences addressing the importance of considering detailed clinical factors for the application of PRS in the prediction of the subsequent events to reflect your consideration to readers as below (line 23, page 14 to line 2, page 15).

“And such a PRS application for the follow-up and treatment possibly will become more effective and precise if the confounding effect of clinical factors such as detailed information of diseases, drugs taking, method of treatment, outcomes, durations through onset, treatment, discharge, and follow-up are considered together.”

 

6. PLOS authors have the option to publish the peer review history of their article (what does this mean?). If published, this will include your full peer review and any attached files.

Do you want your identity to be public for this peer review? For information about this choice, including consent withdrawal, please see our Privacy Policy.

Reviewer #1: No

Reviewer #2: No

---

## [Decision Letter · Decision Letter 1]

5 Jan 2021

PONE-D-20-27402R1

Polygenic risk score validation using Korean genomes of 265 early-onset acute myocardial infarction patients and 636 healthy controls

PLOS ONE

Dear Dr. Bhak,

Thank you for submitting your manuscript to PLOS ONE. After careful consideration, we feel that it has merit but does not fully meet PLOS ONE’s publication criteria as it currently stands. Therefore, we invite you to submit a revised version of the manuscript that addresses the points raised during the review process.

We look forward to receiving your revised manuscript.

Kind regards,

Yiqiang Zhan

Academic Editor

PLOS ONE

Reviewers' comments:

Reviewer's Responses to Questions

**Comments to the Author**

1. If the authors have adequately addressed your comments raised in a previous round of review and you feel that this manuscript is now acceptable for publication, you may indicate that here to bypass the “Comments to the Author” section, enter your conflict of interest statement in the “Confidential to Editor” section, and submit your "Accept" recommendation.

Reviewer #1: All comments have been addressed

Reviewer #2: All comments have been addressed

2. Is the manuscript technically sound, and do the data support the conclusions?

Reviewer #1: (No Response)

Reviewer #2: Partly

3. Has the statistical analysis been performed appropriately and rigorously? 

Reviewer #1: (No Response)

Reviewer #2: N/A

4. Have the authors made all data underlying the findings in their manuscript fully available?

Reviewer #1: (No Response)

Reviewer #2: No

5. Is the manuscript presented in an intelligible fashion and written in standard English?

Reviewer #1: (No Response)

Reviewer #2: Yes

6. Review Comments to the Author

Reviewer #1: (No Response)

Reviewer #2: Manuscript PONE-D-20-27402R1 Thank you for giving us the opportunity to review the manuscript Title: “Polygenic risk score validation using Korean genomes of 265 early-onset acute myocardial infarction patients and 636 healthy controls” We have some points we would like to refer:

- AMI should be subgroup analysis between STEMI and non- STEMI group).

- Death should be identified either cardiac death or all causes of death

- Other MACE components are beneficial in identifying the atherosclerosis progression specialy in young age group as stroke reinfarction or stent thrombosis

7. PLOS authors have the option to publish the peer review history of their article (what does this mean?). If published, this will include your full peer review and any attached files.

Reviewer #1: No

Reviewer #2: No

---

## [Author Response · Author response to Decision Letter 1]

16 Jan 2021

Comments to the Author

1. If the authors have adequately addressed your comments raised in a previous round of review and you feel that this manuscript is now acceptable for publication, you may indicate that here to bypass the “Comments to the Author” section, enter your conflict of interest statement in the “Confidential to Editor” section, and submit your "Accept" recommendation.

Reviewer #1: All comments have been addressed

Reviewer #2: All comments have been addressed

2. Is the manuscript technically sound, and do the data support the conclusions?

Reviewer #1: (No Response)

Reviewer #2: Partly

We have modified the manuscript defining death as all causes of death to accommodate the reviewer's suggestion.

3. Has the statistical analysis been performed appropriately and rigorously?

Reviewer #1: (No Response)

Reviewer #2: N/A

4. Have the authors made all data underlying the findings in their manuscript fully available?

Reviewer #1: (No Response)

Reviewer #2: No

For the data availability issue, we have listed the kind of data available upon request and updated the reason for the restricted data availability with the contact point for the data request in the revised cover letter as below.

“Sequence raw data, individual genotypes, and clinical trait information are under legal restriction in Korea since the data contain potentially identifying or sensitive personal information. Therefore, the above data will be available upon request and after an approval from the Korean Genomics Center’s review board in UNIST. Information about the KGP, present dataset (Cardiomics) and other related data sharing can be found at http://koreangenome.org/Cardiomics.”

5. Is the manuscript presented in an intelligible fashion and written in standard English?

Reviewer #1: (No Response)

Reviewer #2: Yes

6. Review Comments to the Author

Reviewer #1: (No Response)

Reviewer #2: Manuscript PONE-D-20-27402R1 Thank you for giving us the opportunity to review the manuscript Title: “Polygenic risk score validation using Korean genomes of 265 early-onset acute myocardial infarction patients and 636 healthy controls” We have some points we would like to refer:

- AMI should be subgroup analysis between STEMI and non- STEMI group).

- Death should be identified either cardiac death or all causes of death

- Other MACE components are beneficial in identifying the atherosclerosis progression specialy in young age group as stroke reinfarction or stent thrombosis

Thank you for providing us with suggestions. Your point on subgrouping AMI (STEMI and non-STEMI) and outcomes from the patients (MACE, stroke reinfarction, or stent thrombosis) is good which could be easily overlooked in general PRS cardiovascular genomics studies [PMID: 32068818, 30571185, and 30586733]. We have now modified the manuscript by adding a statement that specifies the importance of taking into account of subgroups of disease and kind of outcomes further as below (line 23, page 14 to line 2, page 15).

“And such a PRS application for the follow-up and treatment will possibly become more effective and precise if the confounding effect of clinical factors such as detailed information of diseases, subgroups of disease, drugs taking, method of treatment, kind of outcomes, durations through onset, treatment, discharge, and follow-up are considered together.”

Also, we modified the death denoted in the manuscript as all cause of death (line 1, page 7 in the method section; line 4 to 6, page 11 in the result section).

7. PLOS authors have the option to publish the peer review history of their article (what does this mean?). If published, this will include your full peer review and any attached files.

Do you want your identity to be public for this peer review? For information about this choice, including consent withdrawal, please see our Privacy Policy.

Reviewer #1: No

Reviewer #2: No

---

## [Editor Report · Decision Letter 2]

21 Jan 2021

Polygenic risk score validation using Korean genomes of 265 early-onset acute myocardial infarction patients and 636 healthy controls

PONE-D-20-27402R2

Dear Dr. Bhak,

We’re pleased to inform you that your manuscript has been judged scientifically suitable for publication and will be formally accepted for publication once it meets all outstanding technical requirements.

Kind regards,

Yiqiang Zhan

Academic Editor

PLOS ONE
---

## [Editor Report · Acceptance letter]

25 Jan 2021

PONE-D-20-27402R2 

Polygenic risk score validation using Korean genomes of 265 early-onset acute myocardial infarction patients and 636 healthy controls 

Dear Dr. Bhak:

I'm pleased to inform you that your manuscript has been deemed suitable for publication in PLOS ONE. Congratulations! Your manuscript is now with our production department. 

Kind regards, 

on behalf of

Dr. Yiqiang Zhan 

Academic Editor

PLOS ONE